# Maturation of metabolic connectivity of the adolescent rat brain

**Hongyoon Choi[1,2], Yoori Choi[1], Kyu Wan Kim[1], Hyejin Kang[1], Do Won Hwang[1,2], E Edmund Kim[1,2], June-Key Chung[1], Dong Soo Lee[1,2]\***

[1]Department of Nuclear Medicine, Seoul National University College of Medicine, Seoul, Republic of Korea; [2]Department of Molecular Medicine and Biopharmaceutical Sciences, Graduate School of Convergence Science and Technology, Seoul National University, Seoul, Republic of Korea

**Abstract** Neuroimaging has been used to examine developmental changes of the brain. While PET studies revealed maturation-related changes, maturation of metabolic connectivity of the brain is not yet understood. Here, we show that rat brain metabolism is reconfigured to achieve long-distance connections with higher energy efficiency during maturation. Metabolism increased in anterior cerebrum and decreased in thalamus and cerebellum during maturation. When functional covariance patterns of PET images were examined, metabolic networks including default mode network (DMN) were extracted. Connectivity increased between the anterior and posterior parts of DMN and sensory-motor cortices during maturation. Energy efficiency, a ratio of connectivity strength to metabolism of a region, increased in medial prefrontal and retrosplenial cortices. Our data revealed that metabolic networks mature to increase metabolic connections and establish its efficiency between large-scale spatial components from childhood to early adulthood. Neurodevelopmental diseases might be understood by abnormal reconfiguration of metabolic connectivity and efficiency.

**\*For correspondence:** dsl@snu.ac.kr

**Competing interests:** The authors declare that no competing interests exist.

## Introduction

Recent neuroimaging studies unraveled the developmental changes of the adolescent brains (*Blakemore, 2012*). Among those neuroimaging techniques, positron emission tomography (PET) provides quantitative information regarding regional metabolism or synaptic neurochemicals with high sensitivity. As [18]F-Fluorodeoxyglucose (FDG) is taken up in the brain proportional to cerebral energy consumption and neuronal activity, FDG PET studies have been used to investigate regional energy metabolism during physiologic and pathologic processes. Previous brain developmental studies using FDG PET in humans assumed that the regional increase of metabolism was accompanied by regional increase of neuronal activity during brain maturation (*Chugani et al., 1987*; *Phelps and Mazziotta, 1985*). Despite these previous data, the developmental changes are not understood in terms of dynamic organization of metabolic networks between brain regions.

Since a series of regional brain activity is being coherently organized changing throughout time, sets of specific brain regions are activated or deactivated during resting or various cognitive tasks. These specific regional activity patterns, so-called functional brain networks, are identified by neuroimaging studies (*Deco et al., 2011*). Among several imaging approaches, resting state functional magnetic resonance image (fMRI)-based connectivity analyses revealed maturation of the functional networks during childhood and adolescence (*Fair et al., 2008*; *Fair et al., 2007*; *Supekar et al., 2009*). FDG PET can also reveal functional metabolic network and its maturation, exploiting the coupling between neuronal activity and metabolism. While the fMRI-based functional connectivity measures correlation of fast temporal fluctuations, metabolic connectivity measured by FDG PET reflects

**eLife digest** The brain consumes a great deal of a sugar called glucose, which is delivered to the brain through blood vessels. Active regions of the brain need more glucose, and so the brain has a metabolic network that controls when and where glucose is metabolized. Yet precisely how this metabolic network changes during brain development is not yet understood.

Choi et al. have now monitored the patterns of glucose metabolism in the brains of awake rats as they matured from 'childhood' to early adulthood. The experiments involved injecting the rats with radioactive glucose, and then using a technique called positron emission tomography (commonly known as 'PET scan') to monitor the metabolism of these radioactive sugar molecules in the animals' brains.

Choi et al. showed that the patterns of glucose consumption in the brain shift drastically as the rats mature. Importantly, the findings showed that these shifts in glucose metabolism seem to support the activity of long distance connections that develop as the brain matures. The findings also showed that the increased long distance connections were energy efficient. The results suggest that these metabolic changes are likely a way of maintaining high-energy efficiency that is crucial for the brain to perform normally.

Finally, in addition to revealing the changes involved in normal brain development, these findings may have implications in neurological and psychiatric disorders in which the brain fails to achieve efficient metabolic networks as it matures.

a cumulative energy consumption in several minutes and provides relatively stable information regarding steady resting state (*Choi et al., 2014*; *Di et al., 2012*; *Lee et al., 2008*; *Lee et al., 2012*; *Toussaint et al., 2012*; *Yakushev et al., 2013*). Furthermore, FDG PET measures metabolic activity and connection patterns during awaken states even in animals as in humans, because FDG is taken up in the brain mainly during the period after injection and before the imaging under anesthesia.

To organize highly connected functional brain networks, which can be measured by fMRI or FDG PET, the brain needs efficient energy metabolism (*Bullmore and Sporns, 2012*). As the high energetic costs are required for brain wiring to construct and maintain hub regions with dense connections, hub regions will show high glucose metabolism (*Lord et al., 2013*). Brain regions were found to show different regional energy efficiency so that, for instance, subcortical regions consumed relatively lower metabolic energy per connection (*Tomasi et al., 2013*). Therefore, we can say that metabolism of brain regions could be determined both by the regional complexity of functional networks and their energy efficiency for wiring. Since the developmental changes in functional networks reconfigure the metabolic demands, energy efficiency of the brain networks are going to be changed during maturation.

We primarily aimed to find the maturation-related changes in both regional metabolic activity and metabolic connectivity in adolescent period with a longitudinal study in rats. As the animals allow repeated imaging studies, we could study longitudinal maturation of regional metabolism and connectivity in the same animals. As the small animals also have default mode network (DMN) (*Lu et al., 2012*) defined as brain regions activating during resting state and deactivating during attention-demanding tasks (*Raichle et al., 2001*), specific network components were extracted in the rat brain from minutes-scale metabolic covariance patterns and independent component analysis (ICA). We hypothesized the metabolic networks are dynamically organized during maturation to achieve connection efficiency. We investigated maturation of large-scale metabolic connectivity and analyzed their energy efficiency for wiring in rat brains.

## Results

Voxelwise comparisons were firstly performed among rats aged 5, 10, and 15 weeks to disclose temporal changes of regional metabolic activity. The comparison using statistical parametric mapping revealed that the metabolism increased in 10-week-old age in bilateral frontal cortices and anterior aspect of striatum compared with 5-week-old age and decreased in bilateral cerebellar cortices, thalamus, parieto-occipital cortices, and retrosplenial cortices (*Figure 1A*) (representative FDG PET

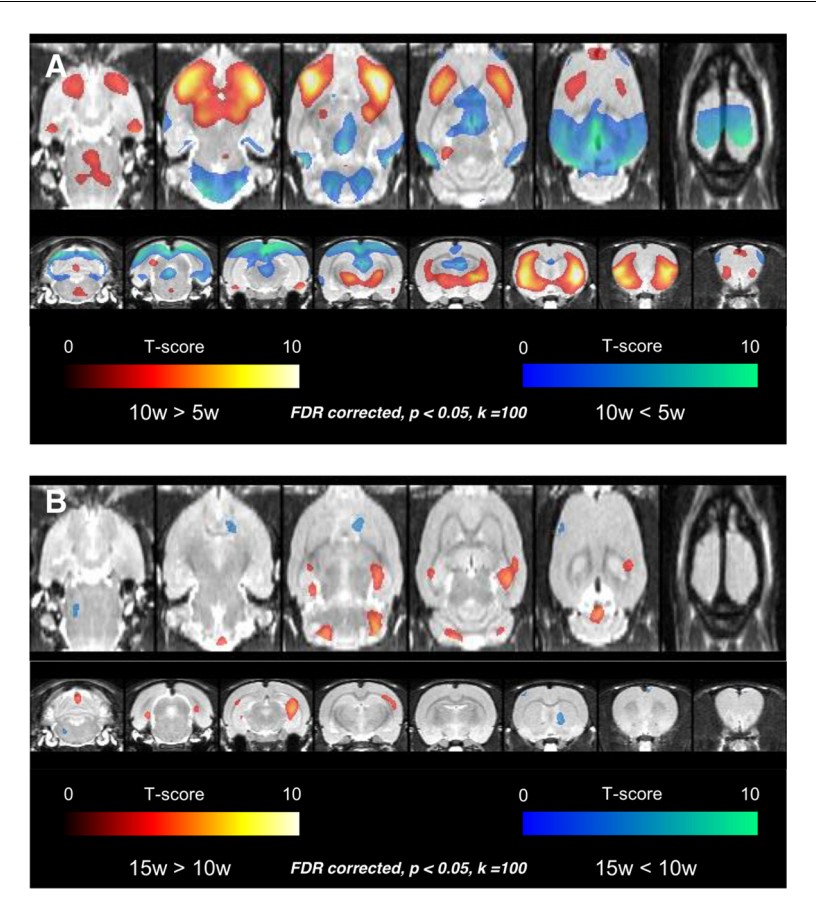

**Figure 1.** Voxelwise comparison results among rats aged 5, 10, and 15 weeks. (**A**) Age-related increase of metabolism was found in bilateral frontal cortices and anterior aspect of striatum in 10 weeks. Age-related decrease of metabolism was detected in bilateral cerebellar cortices, thalamus, parietooccipital cortices, and retrosplenial cortices. (**B**) In the comparison between 10-week and 15-week-old rats, age-related increase of metabolism was found in the clusters of both hippocampi and decrease of metabolism in the small clusters of right striatum and left frontal cortex. The finding implies that considerable metabolic maturation in frontal cortices occurs between 5 and 10 weeks, *i.e.* during adolescence.

The following figure supplements are available for figure 1:

**Figure supplement 1.** Representative FDG PET images for each age.

**Figure supplement 2.** Voxelwise comparison between 5-week-old and 15-week-old rats.

images for each age are shown in *Figure 1—figure supplement 1*). Again, metabolism increased in 15-week-old age compared with 10-week-old age, in the clusters of both hippocampi and decreased in the small clusters of right striatum and left frontal cortex (*Figure 1B*, *Figure1— figure supplement 2*).

Interregional metabolic correlation on FDG PET was supposed to reflect interregional covariance patterns of neuronal activities. An ICA was performed to yield regional contributing components of metabolic networks. A total of 13 metabolic independent components (ICs) were identified in rats similar to components found in previous reports in humans on FDG PET (*Di et al., 2012*; *Toussaint et al., 2012*; *Yakushev et al., 2013*). All ICs were displayed in *Figure 2—figure supplement 1*, and a threshold z > 1.5 was applied for the voxels for display purposes. Four particular components, IC1, IC5, IC8, and IC9, were selected (*Figure 2*), which anatomically corresponded to previously alleged limbic/anterior DMN (IC1), posterior DMN (IC5), motor (IC8), and somatosensory

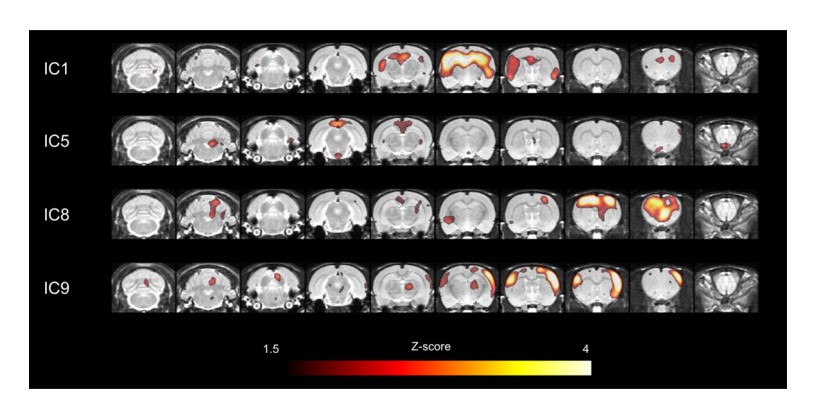

**Figure 2.** Major network components in cerebral cortices obtained by ICA. Among 13 large-scale network components, 4 independent components which corresponded to those in humans were selected for further connectivity analyses. In particular, independent component (IC) 1 included bilateral hippocampi and medial prefrontal cortex. IC5 included retrosplenial cortex, a core of default-mode network. IC8 included motor cortex and IC9 included somatosensory cortex. ICA, independent component analysis.

The following figure supplements are available for figure 2:

**Figure supplement 1.** Independent components identified by ICA of FDG PET images (n = 88).

**Figure supplement 2.** Eight volume-of-interests (VOIs) were defined as spheres with a radius of 0.8 mm (actual size), located on each of independent components (ICs).

network (IC9), respectively (*Lu et al., 2012*). IC1 included the clusters of dorsal hippocampi and medial prefrontal cortex. IC5 included the retrosplenial cortex, known as a hub of posterior DMN in rats (*Lu et al., 2012*). It is anatomically adjacent to the posterior cingulate and precuneus in human as cores of DMN (*Raichle et al., 2001*). These were used further to identify maturing patterns of metabolic networks consisting of IC-derived volume-of-interests (VOIs). Eight VOIs (3 for IC1, 1 for IC5, 2 for IC8 and IC9, respectively) were selected taking anatomical structures into consideration (*Figure 2—figure supplement 2*) (stereotaxic coordinates and abbreviations of VOIs are summarized in *Table 1*).

Interregional correlations were calculated between paired VOIs to yield 28 pairs (28 edges upon 8 nodes). This VOI-based metabolic interregional correlation was assumed to yield functionally coherent covariance patterns among spatial components to represent metabolic connectivity. Metabolic connectivity obtained this way was used to investigate the changes of metabolic networks during maturation. *Figure 3A* shows interregional correlation representing connection strength of pairs of VOIs in rats aged 5, 10, and 15 weeks. Significantly different connections in interregional correlation were examined using nonparametric permutation tests (*Figure 3—figure supplement 1*) using pseudorandom relabeling 5-week/10-week-old rat images or 10-week/15-week-old ones, followed by FDR correction for multiple comparisons (*Kim et al., 2015*). Connectivity matrices (*Figure 3A*) and p-values for the difference in connection strength (*Figure 3—figure supplement 2*) between paired groups suggested that the connection between pairs of retrosplenial, medial prefrontal, and sensorimotor cortices was strengthened from 5 to 10 weeks, while connectivity involving limbic regions did not change. According to aging, pairs of anterior-posterior connections were significantly strengthened when comparing 10-week-old rats with 5-week-old rats (*Figure 3B*). The increase became prominent when we compared 15-week-old rats from 5-week-old rats. Connectivity was not significantly weakened between areas during maturation (*Figure 3—figure supplement 2*).

We further investigated energy efficiency to configure the metabolic connectivity between spatial components at each period of maturation. The energy efficiency was defined as the ratio of metabolic connection strength, a sum of weights of links connected to each VOI, to normalized FDG uptake of each VOI. Significantly different metabolic energy efficiency was also examined using the permutation test (*Figure 3—figure supplement 1*). In 5-week-old rats, energy efficiency was

**Table 1.** List of the streotaxic coordinates for the functional networks nodes in rat brain.

| | Abbreviation | Paxinos atlas (mm) | | |
| --- | --- | --- | --- | --- |
| | | ML | DV | AP |
| IC1 | | | | |
| Left Hippocampus | Hp_L | -2.6 | 3.2 | -3.2 |
| Right Hippocampus | Hp_R | 2.6 | 3.2 | -3.2 |
| Medial Prefrontal Cotex | MedF | 0.0 | 3.8 | 4 |
| IC5 | | | | |
| Retrosplenial Cortex | Rsp | 0.0 | 1.8 | -6.4 |
| IC8 | | | | |
| Left motor cortex | Mot_L | -2.2 | 2.2 | 2.2 |
| Right motor cortex | Mot_R | 2.2 | 2.2 | 2.2 |
| IC9 | | | | |
| Left somatosensory cortex | SS_L | -5.0 | 3.8 | 0.6 |
| Right somatosensory cortex | SS_R | 5.0 | 3.8 | 0.6 |

ML: medial-lateral; AP: anterior-posterior; DV: dorsal-ventral; IC: independent component.

relatively lower in midline structures, that is, medial prefrontal and retrosplenial cortices than sensori-motor cortices. In 10- and 15-week-old age compared with 5-week-old age, energy efficiency of the midline structures significantly increased almost reaching those of the sensorimotor cortices (*Figure 4*).

## Discussion

In the present study, large-scale brain network based on glucose metabolic usage measured by PET was analyzed to evaluate the maturation of the metabolic brain connectivity in rats. We identified the specific pattern of changes in regional metabolism and metabolic connectivity during brain development. In adolescent (10 weeks) and early-adult (15 weeks) period, metabolism increased in the bilateral frontal lobes compared to childhood (5 weeks), whereas metabolism decreased in the posterior cerebrum, thalamus, and cerebellum. Metabolic network components extracted using ICA were default-mode, limbic, motor, and somatosensory ones, which have been regarded as important functional components also in humans. Further connectivity analyses using VOIs put on these components showed connection strength increased particularly in the anterior-posterior connections during maturation. This connection strength increase was accompanied by increased energy efficiency in the midline structures including medial prefrontal and retrosplenial cortices. Our work suggested large-scale functional networks matured to increase anterior-posterior long-distance connections and achieved energetically efficient wiring in the midline structures during maturation from childhood via adolescence to early adulthood.

Voxelwise comparison revealed metabolism increase in bilateral frontal cortices during maturation, prominent during 510 weeks of ages, which complied with previous perfusion studies. A study using [99m]Tc-HMPAO SPECT showed perfusion increase in the neocortex at juvenile to young adult periods in mice, particularly in frontal lobes relative to the cerebellum and subcortical structures (*Apostolova et al., 2012*). In humans, perfusion was lower in the neonates' neocortex while higher in the basal ganglia and cerebellum (*Fockele et al., 1990*). Additionally, positive correlation was found between age and frontal lobe perfusion (*Kuji et al., 1999*). FDG PET studies in human showed significantly higher metabolism mostly in anterior cingulate and thalamus before 25-year-old (*Van Bogaert et al., 1998*) and linear metabolism increase from the age of 115 in prefrontal/orbito-frontal cortices (*Kang et al., 2004*). As a first study in terms of longitudinal age-related brain metabolism in rats, our findings of age-related frontal metabolism increase corresponded to human PET findings as well as perfusion studies in rodents.

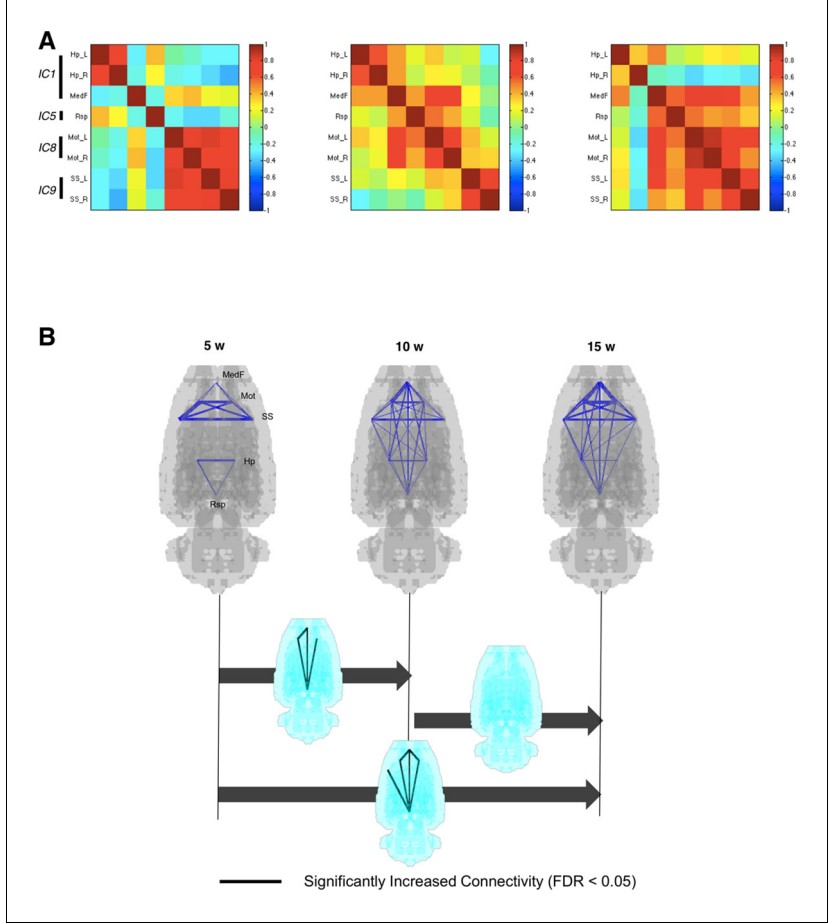

**Figure 3.** Metabolic connectivity between brain regions. (**A**) To find functional relevance between brain regions, metabolic activities of several brain regions were correlated to each other. Eight volume-of-interests (VOIs) were selected from the results of independent component analysis, and interregional correlations between them were calculated. Correlation strengths between different networks increased according to age. (**B**) Blue lines indicate pairwise connections which showed positive correlation. Note that line width means strength of the connectivity. Significant increase of metabolic connectivity is shown mainly in anterior-posterior connections according to maturation (for multiple comparison, false discovery rate < 0.05 was applied). More number of pairs of significant increase in metabolic connectivity was shown between 5 and 15 weeks of age than 5 and 10 weeks of age.

The following figure supplements are available for figure 3:

**Figure supplement 1.** Statistical analysis for metabolic connectivity maturation based on permutation test.

**Figure supplement 2.** Statistical significance (p-values) for difference in strengths of connections between groups of different aged rats.

Analysis of metabolic network are based on the sequential coupling of neuronal activity, brain metabolism, and blood flow, which more closely represent the neuronal networks than blood flow networks. Because metabolic activity measured by FDG uptake reflects the cumulative energy consumption in steady states, while BOLD signal from fMRI reflects fast temporal fluctuation of physiologic factors such as blood flow, blood oxygenation, and cerebral metabolic rate of oxygen, the metabolic network provides distinct information from the functional network on fMRI as well as from the structural network disclosed on diffusion tensor imaging (*Di et al., 2012*; *Wehrl et al., 2013*). Furthermore, network construction using neuronal activity-coupled metabolism found on FDG PET could take regional metabolic activities into account to disclose energy efficiency rather than only

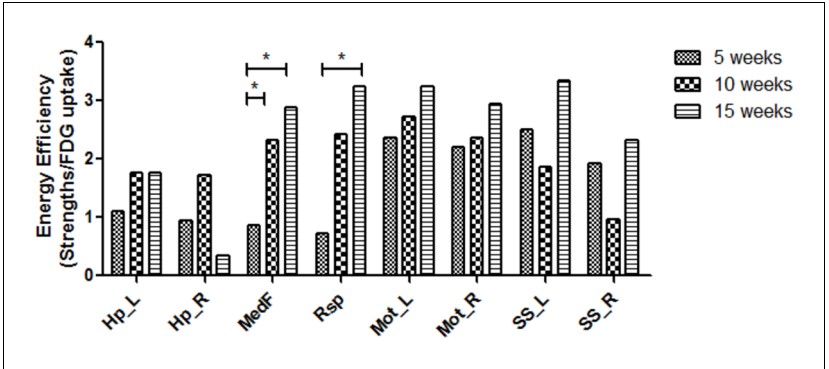

**Figure 4.** Increased energy efficiency in the medial prefrontal and retrosplenial cortices during maturation. Energy efficiency defined as a ratio of metabolic connectivity strength to normalized FDG uptake was estimated for each region. The energy efficiency changes and was interpreted to be reorganized during brain maturation. While efficiency in the midline structure was lower at 5 week of age, that is, medial prefrontal and retrosplenial cortices, but increased significantly in 10 and 15 weeks. (*p < 0.05 based on Bonferroni correction over permutation test). FDG, [18]F-Fluorodeoxyglucose.

The following source data is available for figure 4:

**Source data 1.** Energy efficiency of each brain region.

concentrate on the changes of fluctuating neural activities without considering absolute amount of regional perfusion or metabolism (*Hyder et al., 2011*; *Smith et al., 2002*).

Using metabolic network analyses, we could identify networks of rat brains similar to humans using ICA. In human brains, ICA for metabolic networks revealed DMN successfully (*Toussaint et al., 2012*; *Yakushev et al., 2013*). We also found homologous patterns in rats to those of humans. According to the studies comparing the networks acquired from fMRI and PET, each DMN from these modalities shared quite the similar anatomical regions, although distinctive patterns were found on these two modalities (*Di et al., 2012*; *Wehrl et al., 2013*). Of note, our PET-derived IC5 shared the regions of fMRI-derived DMN including retrosplenial cortices and small clusters of bilateral posterior hippocampi. The IC1 shared the regions of anterior part of DMN derived from fMRI, medial frontal cortices (*Lu et al., 2012*). Considering that DMN-related regions are deactivated during attention-demanding and working memory tasks (*Raichle et al., 2001*), DMN may be constructed during brain maturation in adolescence to learn attention-demanding tasks. Previous human studies on the fMRI-derived functional networks revealed connectivity increase in DMN regions during maturation, particularly between the anterior and posterior midline structures (*Fair et al., 2008*), which might be closely associated with the developmental process of working memory (*Satterthwaite et al., 2013*). Moreover, fMRI-derived DMN showed two distinctive clusters, temporal-prefrontal and parietal subsystems, which considerably corresponded to our IC1 and IC5 (*Lu et al., 2012*). According to our results, the metabolic connections of these two clusters of DMN developed during brain maturation in rats. In spite of homologous network patterns of rat brain, metabolic networks derived from FDG PET were partly different from fMRI-derived networks. A discrepancy in DMN between fMRI and PET was also reported in human (*Di et al., 2012*). In PET-derived networks in the rat brain, anterior and posterior part of DMN was separately extracted (IC1 and IC5). Furthermore, cingulate cortex was rarely included in IC1 or IC5, which was different from previous fMRI-derived DMN (*Lu et al., 2012*; *Sierakowiak et al., 2015*). Nevertheless, a hub region of posterior part of rat DMN was retrosplenial cortex and that of anterior part was medial prefrontal cortex in fMRI study, which corresponded to PET-derived IC5 and IC1, respectively.

During connectivity maturation, metabolic connections between component VOIs strengthened, particularly in anterior-posterior long-distance ones. Along with increased long-distance connections, the retrosplenial and medial prefrontal cortices showed efficient energy consumption. This maturation pattern may represent enhancement of efficiency of information flow between network components, which has been suggested in human brain maturation (*Fair et al., 2007*; *Stevens et al.,*

*2009*). The strengthened connections between long-distance network components are likely to support dynamically organizing brain networks to achieve efficient connections according to cognitive demands (*Bullmore and Sporns, 2012*). The functional interactions of large-scale brain networks are dynamically evolving, which could be associated with cognitive, behavioral developments during maturation and pathophysiology of brain disorders (*Deco et al., 2011*). In this investigation using metabolic connectivity analysis, we could localize the components, that is, medial prefrontal and retrosplenial areas, showing strengthened connections and at the same time enhanced connection efficiencies.

Cognitive developments demand highly connected functional brain networks, particularly in hub regions (*Micheloyannis et al., 2009*; *van den Heuvel and Sporns, 2013*). They could accompany high-energy consumption in spite of limited supply of glucose for brain (*Smith et al., 2002*; *Tomasi et al., 2013*). Our results of increased energy efficiency in retrosplenial and medial prefrontal cortex imply long-distance connectivity maturation coincide with the redistribution of energy consumption. High-energy efficiency in hubs was previously reported in adult human brain (*Tomasi et al., 2013*), which was recapitulated in our study in rats, and we also found that their high efficiency developed during brain maturation. Comprehensively, large-scale networks are formed during adolescent period accompanied by these networks' efficient energy consumption for long-distance connectivity. As deficits in these relationships are found in several brain disorders (*Broyd et al., 2009*), metabolic network analyses might be used to find the relevant functional connectivity abnormalities. Specifically, the hub regions are well known as culprit components for several developmental and degenerative brain disorders (*Fornito et al., 2015*). Our results could explain the underlying biological background of the vulnerability that hub regions might be easily influenced by developmental problems in reconfiguration of metabolic connectivity efficiency.

To find the disease-related abnormalities differentiated from normal developmental changes in neurological or psychiatric disorders, we need to characterize first the normal developmental patterns of metabolic networks. Although metabolic networks have advantages to reveal steady-state connectivity in longer term scale than resting fMRI, it is difficult to perform longitudinally repeated PET in the growing normal children. And thus, our findings in rats of maturational changes of metabolic connectivity can be referred to as surrogates, considering the homology between humans and rats in network components disclosed by ICA (*Lu et al., 2012*; *Wehrl et al., 2013*).

Although rat brains have similar features of metabolic network to human brains, there are several technical issues of interest. Unlike human studies, rats should be anesthetized during either FDG injection or image acquisition. Since anesthesia could affect brain metabolism, rats were awake after the injection until imaging to minimize the anesthesia effects. The images acquired from small animal PET had the voxel size of 0.3875 mm and physical spatial resolution of PET scanner is more than 1 mm due to positron range and acollinearity. Thus, we were afraid that it might prevent the delineation of network components during ICA. However, despite this limited spatial resolution, the brain regions with specific covariance patterns were identified and further metabolic connectivity analyses were finally feasible to provide characteristic pattern of brain maturation.

## Conclusion

We showed that metabolic activity of rat brain matured to build connectivity between network components accompanied by enhanced energy efficiency from childhood to early adulthood. Metabolism increased during adolescence in frontal cortices compared to childhood in rat brain similarly to that in human brain. Components of metabolic networks were identified, and connectivity analysis showed efficient connection developed between the components particularly pair of VOIs including DMN during adolescent period. Furthermore, the midline structures showed increase in energy efficiency of metabolic connections. This study about metabolic network maturation gave insights into normal brain developments and might elucidate the plausible pathophysiology of neuropsychiatric developmental diseases.

## Materials and methods

### FDG PET for animals

Thirty adult male Sprague-Dawley (SD) rats (Koatech, Seoul, Korea) were used for FDG PET scans. They were kept at standard laboratory condition (22–24°C, 12 hr light and dark cycle) with free access to water and standard feed. All the experimental procedures were approved by Institutional Animal Care and Use Committee at Seoul National University Hospital (IACUC Number 13–0224). FDG PET images were acquired at the age of 5, 10, and 15 weeks. PET scans were performed on a dedicated small animal PET/CT scanner (eXplore VISTA, GE Healthcare, WI). Rats were fasted for at least 8 hr before the start of the study. Rats were anesthetized 2% isoflurane at 1–1.5 L/min oxygen flow for 5–10 min. After an intravenous bolus injection (0.3–0.5 mL/rat) of FDG (100–150 MBq/kg), each rat has 45 min period of FDG uptake (*Schiffer et al., 2007*). Static scans at 45 min after the injection best reflects absolute rates of glucose metabolism in rodents (*Schiffer et al., 2007*). During FDG uptake period, rats were awake for 35 min and anesthetized for preparation, 10 min before PET/CT scans. Emission scan data were acquired for 20 min with the energy window 400–700 keV and reconstructed by a three-dimensional ordered-subsets expectation maximum (OSEM) algorithm with attenuation, random and scatter correction. The voxel size was $0.3875 \times 0.3875 \times 0.775$ mm. We acquired 88 PET images considering voxelwise statistical difference after multiple comparison correction: 30 scans at 5 weeks, 30 scans at 10 weeks, and 28 scans at 15 weeks.

### Image preprocessing

For preprocessing, all voxels were scaled by a factor of 10 in each dimension. All brain PET images were spatially normalized to the FDG rat brain template (*Schiffer et al., 2006*) (PMOD 3.4, PMOD group, Zurich, Switzerland) using nonlinear registration on Statistical Parametric Mapping (SPM8, University College of London, London, UK). After spatial normalization, a binary mask for brain was applied. The images were smoothened by Gaussian filter of 12 mm full width at half maximum. For scaling voxel intensities, the voxel counts were normalized to the global brain uptake in each PET image.

### Voxelwise comparison of regional FDG uptake along the brain maturation

Testing for age effects on the brain metabolism was performed with SPM. Paired t-test was performed between two groups of PET images acquired at 5 and 10 weeks. Additional paired t-tests were performed between PET images at 10 and 15 weeks and PET images at 5 and 15 weeks. False discovery rate (FDR) corrected $p < 0.05$ was set as the significance threshold and an extent threshold of 100 contiguous voxels was applied.

### Identification of network components using ICA

We applied a group ICA algorithm to define coherent network components (GIFT, http://mialab.mrn.org/, GIFT ver 2.0a). All preprocessed PET images (n = 88) were included in this spatial ICA to find spatially independent components and coherently activated regions among subjects. The group ICA approaches have been used to obtain multivariate patterns for metabolic networks (*Toussaint et al., 2012*; *Yakushev et al., 2013*). Prior to perform ICA, the optimal number of components extracted from PET images was determined. We used the dimensional estimation algorithm implemented in GIFT software based on the assessment of entropy rate of independent and identically distributed (*i.i.d.*) Gaussian random process (*Li et al., 2007*; *Xu et al., 2009*). The estimated optimal number was thirteen components. ICA was performed using an infomax neural network algorithm that minimized the mutual information of the outputs. The resulting independent components were z-transformed and visualized using the threshold $z > 1.5$.

### Connectivity analysis using independent components as nodes to reveal the changes during maturation

To compare the changes of brain metabolic connectivity according to age, we engaged VOI-based correlation analyses. VOIs were placed on four ICs (IC1, IC5, IC8, and IC9) referring to rat brain atlas, which was constructed on 3D digital map based on Paxinos and Watson atlas (*Schiffer et al., 2006*;

*Toga et al., 1995*). We selected these ICs to concentrate on the relationship between functional components including DMN as intrinsically coherent regions at resting state and sensory-motor networks. The total eight VOIs (3 on IC1, 1 on IC5, 2 on IC8 and IC9) were defined as spheres with a radius of 8 mm, centered at each IC (the actual size of a radius was 0.8 mm as voxels were scaled by a factor of 10). The size of VOI was determined considering initial voxel size (0.775 mm) and to a representative small cluster of a network not to overlap other networks. The VOIs were displayed in *Figure 2—figure supplement 2*. We compared the connectivity between rats aged 5, 10, and 15 weeks.

We obtained normalized FDG uptake in the VOIs of each rat and Pearson's correlation coefficients were calculated between the pairwise VOIs (8x7/2= 28 paired VOIs) using subjects variation. These correlation coefficient matrices were constructed for 5 week-, 10 week-, and 15 week-old rats. For statistical comparison of correlation matrices between these groups of different ages, we performed permutation tests. We tested whether there was significantly different connectivity, represented by Fisher-transformed Z values, between 5 vs 10 weeks and 10 vs 15 weeks of age. At first, PET images of each age group were randomly permuted to make pseudo-random groups reassigned 10,000 times and from each paired group of rats, interregional correlation matrices were calculated. Type I errors were determined by the comparison between the observed Z score for each connection of VOI pairs and Z score distribution of VOI pairs from the permuted data (*Figure 3—figure supplement 1*). For multiple comparison correction, we applied false-discovery rate (FDR) at a threshold of FDR < 0.05.

## Maturation of metabolic energy efficiency

To analyze energy efficiency for metabolic connectivity, undirected networks with the eight nodes were constructed where strength of each connection was simply defined as correlation coefficients. Strength of metabolic connectivity was calculated for each VOI as a sum of weights of positive links (correlation coefficients) per VOI (*Kaiser, 2011*). The ratio of metabolic connectivity strength to normalized metabolic activity was defined as energy efficiency for each VOI (*Tomasi et al., 2013*). For statistical comparison of energy efficiency of VOIs between 5, 10, and 15 week-old rats, we performed permutation tests. Again PET images of each paired group of rats were permuted to make pseudo-random groups reassigned 10,000 times and distribution of metabolic energy efficiency of each VOI was drawn. The observed energy efficiency was compared with distribution of energy efficiency of each of VOIs from the permuted data (*Figure 3—figure supplement 1*).

## Acknowledgements

This research was supported by a grant of the Korea Health Technology R&D Project through the Korea Health Industry Development Institute (KHIDI), funded by the Ministry of Health & Welfare, Republic of Korea (HI14C0466), and funded by the Ministry of Health & Welfare, Republic of Korea (HI14C3344), and funded by the Ministry of Health & Welfare, Republic of Korea (HI14C1277), and a grant of the Korean Health Technology R&D Project, Ministry of Health & Welfare, Republic of Korea (HI13C1299). This research was supported by the Original Technology Research Program for Brain Science through the National Research Foundation of Korea (NRF) funded by the Ministry of Education, Science and Technology (2015028926).

## Additional information

### Funding

| Funder | Grant reference number | Author |
|---|---|---|
| Ministry of Health and Welfare | HI14C0466 | Dong Soo Lee |
| Ministry of Health and Welfare | HI14C3344 | Dong Soo Lee |
| Ministry of Health and Welfare | HI14C1277 | Dong Soo Lee |
| Ministry of Health and Welfare | HI13C1299 | Dong Soo Lee |
| Ministry of Education | 2015028926 | Dong Soo Lee |

The funders had no role in study design, data collection and interpretation, or the decision to submit the work for publication.

## Author contributions

HC, Acquisition of data, Analysis and interpretation of data, Drafting or revising the article; YC, Acquisition of data, Analysis and interpretation of data; KWK, Acquisition of data; HK, DWH, EEK, JKC, Analysis and interpretation of data; DSL, Conception and design, Drafting or revising the article

## Ethics

Animal experimentation: All the experimental procedures were approved by Institutional Animal Care and Use Committee at Seoul National University Hospital (IACUC Number 13-0224).

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
