## [Decision Letter]

Thank you for submitting your work entitled "Maturation of Metabolic Connectivity of the Adolescent Rat Brain" for consideration by *eLife*. Your article has been reviewed by two peer reviewers, and the evaluation has been overseen by David Kleinfeld as Reviewing Editor and Timothy Behrens as the Senior Editor.

The reviewers have discussed the reviews with one another and the Reviewing Editor has drafted this decision to help you prepare a revised submission.

Both reviewers agree that this work adds new information on the normal brain development of metabolic activity that might have relevance for neuropsychiatric disorders. The studies would not have been possible to conduct in humans. Yet some concerns remain to be addressed.

Please explain how the large size total VOI (8 mmm radius) affects the spatial certainty for the results.

While you argue that you are studying the default mode network (DMN), some of the core regions of the DFM, the posterior cingulate cortex and the precuneus, are not included. Please clarify this point.

You note in the Results section that independent component (IC) five includes the "retrosplenial cortex, anatomically adjacent to the posterior cingulate and precuneus in human as cores of DMN". Other studies in rats using even larger voxels (0.5 mm-cubed) have defined the cingulate cortex (Sireakowiak et al., PLoS One, 2015). Please explain why you don’t use IC in the DFM when you aim to study that network.

All of these changes will clarify and improve your manuscript. None of these changes require additional data to be obtained.

*Reviewer #1:*

In this paper, FDG-PET was used to define metabolically active networks in the developing rat brain. They examined totally 30 rats at 5, 10 and 15 weeks of age and used independent component analysis to define the nodes in the networks. VOI-based correlation analyses was used to compare the changes of brain metabolic connectivity according to age.

In 5 week-old rats, energy efficiency was lower in midline structures (medial prefrontal and retrosplenial cortices) than in sensorimotor cortices. At 10 and 15 weeks-of-age, energy efficiency of the midline structures (here defined as components of the default mode network) almost reached those of the sensorimotor cortices. The authors conclude that metabolic activity in the rat brain matured to build connectivity between network components accompanied by enhanced energy efficiency from childhood to early adulthood.

This study combines FDG PET and ICA to study a new topic. The results are in line with previous fMRI results showing in humans showing that sensorimotor networks develop before default mode network and other higher order networks. The results add new information on the normal brain development of metabolic activity that might have relevance for neuropsychiatric disorders. The studies would not have been possible to conduct in humans, as correctly argued in the manuscript.

1) I would like the authors to explain how the large size total VOI (8 mmm radius) affects the spatial certainty for the results, when it comes to certain networks.

2) Along the same line, my major concern is that the authors argue that they study the default mode network, but actually some of the core regions of the default mode network, the posterior cingulate cortex and the precuneus, are not included. The authors state in the Results section that independent component 5 included the "retrosplenial cortex, anatomically adjacent to the posterior cingulate and precuneus in human as cores of DMN". Other studies in rats using even larger voxels (0.5 mm^3^) have defined the cingulate cortex (Sireakowiak et al., 2015 for example). I would like the authors to explain why they don’t use IC in the DFM when they aim to study that network.

*Reviewer #2:*

This paper studies the interesting effect of metabolic information derived from FDG-PET imaging data in rats. Rats (n=30) were followed for 15 weeks and scanned in 5 week intervals. The authors show a reconfiguration of brain networks, derived from static PET imaging data. Metabolic connectivity mapping (Cometomics) is a very new technique, and no firm standards have been established so far. The applied methodology as well as animal number is appropriate. The results, based on ICA and graph based analysis are reasonable. The Discussion is also reasonable and covers many different aspects, albeit it could be more detailed especially regarding the possible origins of metabolic connectivity. In conclusion, this article is a valuable contribution to the field of brain connectivity mapping, especially since it uses metabolic information.

---

## [Author Response]

Reviewer #1:

*1) I would like the authors to explain how the large size total VOI (8 mmm radius) affects the spatial certainty for the results, when it comes to certain networks.*

We appreciate this reviewer’s comment with regard to the size of VOI. As images were rescaled (10 times) during initial preprocessing step for the further spatial registration process on SPM, the actual size of VOI was 0.8 mm. For rescaling process, only image header information was changed. We corrected some sentences to clarify the real size of VOIs.

The size was small to represent each network and anatomical region and not to overlap different networks. Furthermore, since initial voxel size was 0.3875 x 0.3875 x 0.775, we selected the spherical VOI with 0.8 mm radius which is larger than an initial voxel. We described this in the manuscript, which now reads:

“The total 8 VOIs (3 on IC1, 1 on IC5, 2 on IC8 and IC9) were defined as spheres with a radius of 8 mm, centered at each IC (The actual size of a radius was 0.8 mm as voxels were scaled by a factor of 10). The size of VOI was determined considering initial voxel size (0.775 mm) and a representative small cluster of a network not to overlap other networks." (Materials and methods)

“Eight volume-of-interests (VOIs) were defined as spheres with a radius of 0.8 mm (actual size), located on each of independent components (IC).” (Figure 2—figure supplement 2 legend)

*2) Along the same line, my major concern is that the authors argue that they study the default mode network, but actually some of the core regions of the default mode network, the posterior cingulate cortex and the precuneus, are not included. The authors state in the Results section that independent component 5 included the "retrosplenial cortex, anatomically adjacent to the posterior cingulate and precuneus in human as cores of DMN". Other studies in rats using even larger voxels (0.5 mm^3^) have defined the cingulate cortex (Sireakowiak et al., 2015 for example). I would like the authors to explain why they don’t use IC in the DFM when they aim to study that network.*

We thank the reviewer for pointing out an important issue with regard to the default mode network in this study. Some previous studies tried to extract DMN using fMRI in a rodent brain, which extracted midline structures including medial frontal, cingulate, retrosplenial cortex and parts of hippocampus. Sireakowiak et al. (Plos One, 2015) used a seed placed on cingulate to extract DMN which naturally include cingulate cortex. Rat DMN derived from fMRI using ICA (Lu et al., PNAS, 2012) extracted quite similar structures. The ICA derived result revealed that hub regions were retrosplenial cortex (for posterior DMN) and medial prefrontal cortex (for anterior DMN), which corresponded to our ICs. Because of the results of DMN hubs in rats, task-related network patterns have been studied using these two regions (for example, Jennifer Li, et al., J Neurosci., 2015), retrosplenial and medial frontal cortices regarded as cores of DMN. The previous reports implied that these two hub regions in the rodent brain could functionally correspond to hubs of DMN in human brain, posterior cingulate cortex/precuneus and anterior medial frontal cortex. Therefore, in our study, ICs which include these two anatomical regions were considered for further analysis.

Although PET-based ICs shared regions with previously reported fMRI-based networks, discrepancy was also found in our study. This discrepancy has been also reported in human. However, physiological background of this discrepancy is still unknown. According to the reviewer’s comment, we discussed these issues seriously in the manuscript, which now reads:

“Of note, our PET-derived IC5 shared the regions of fMRI-derived DMN including retrosplenial cortices and small clusters of bilateral posterior hippocampi. The IC1 shared the regions of anterior part of DMN derived from fMRI, medial frontal cortices.” (Discussion, fourth paragraph)

“In spite of homologous network patterns of rat brain, metabolic networks derived from FDG PET were partly different from fMRI-derived networks. […] Nevertheless, a hub region of posterior part of rat DMN was retrosplenial cortex and that of anterior part was medial prefrontal cortex in fMRI study, which corresponded to PET-derived IC5 and IC1, respectively.” (Discussion, same paragraph)

“IC5 included the retrosplenial cortex, known as a hub of posterior DMN in rats (Lu et al., 2012). It is anatomically adjacent to the posterior cingulate and precuneus in human as cores of DMN (Raichle et al., 2001).” (Results)